# Multifunctional Optical Vortex Beam Generator via Cross-Phase Based on Metasurface

**DOI:** 10.3390/nano12040653

**Published:** 2022-02-15

**Authors:** Kuangling Guo, Yue Liu, Li Chen, Zhongchao Wei, Hongzhan Liu

**Affiliations:** Guangdong Provincial Key Laboratory of Nanophotonic Functional Materials and Devices, School for Information and Optoelectronic Science and Engineering, South China Normal University, Guangzhou 510006, China; klguo99@163.com (K.G.); liuyyy_chn@163.com (Y.L.); 2019022102@m.scnu.edu.cn (L.C.); wzc@scnu.edu.cn (Z.W.)

**Keywords:** multifunctional metasurfaces, optical vortex, cross-phase

## Abstract

We propose a multifunctional optical vortex beam (OVB) generator via cross-phase based on a metasurface. Accordingly, we separately investigate the two different propagation characteristics of OVB modulated by the low-order cross-phase (LOCP) and the high-order cross-phase (HOCP) in a self-selected area. When LOCP modulation is added to OVB, topological charges can be measured for any order of OVB. Moreover, we achieve the rotation tunable performance successfully by adding the rotation component. Then, we realize the function of polygonal beam generation and singularities regulation with the HOCP. The order of the HOCP is exactly equal to the number of a polygon OVB’s sides. The waist radius and usable width of the beam lengthens as the distance of the self-selected area increases. When the conversion rate is doubled, the distance between singularities widens by about 0.5 μm. The proposed OVB generator provides a simple strategy for detecting the value of topological charges and achieving OVB shaping and singularity manipulation simultaneously. We hope this can open new horizons for promoting the development of photon manipulation, optical communication, and vortex beam modulation.

## 1. Introduction

As early as 1992, Allen et al. [1] first discovered that in addition to the general spin angular momentum (SAM) [2], photons can also carry orbital angular momentum (OAM). Optical vortex beam (OVB), with the phase factor exp(i*lφ*), is a special kind of beam carrying OAM of *lħ* per photon, where *l* denotes the topological charges (TCs), which can be any integer, *φ* denotes the azimuthal angle, and *ħ* is the Planck constant. Due to the particularity of carrying OAM, more and more scholars started to study the generation of the OVB. As the research progresses, it is found that because of the arbitrary value of *l*, high-dimensional OAM contains unlimited eigenstates. Thus, the OAM carried by a single photon can theoretically carry an infinite amount of information [3]. In this way, since the OVB can extend the communication capability through strictly orthogonal channels of different OAM modes, it can be greatly improved the information carrying capacity of optical fiber communication systems [4,5,6], thereby ameliorating the influence of propagation distance and capacity on the development of optical fiber communication systems. The OVB also plays an important role in optical communication [7,8], quantum information processing [9], and nonlinear optics [10,11,12]. Recently, the traditional OVB generation methods that have been proposed include spiral phase plates [13,14], antenna arrays [15], sub-wavelength gratings [16,17], and computer holograms [18]. However, the above methods greatly limit the practical applications of OVB due to complex structure or low efficiency. Therefore, scholars have turned their attention to the direction of reducing the volume of the structure.

Metamaterial, formed by artificial periodic microstructured materials, has attracted much interest in the past few decades and produced groundbreaking electromagnetic and photonic phenomena [19,20,21]. The two-dimensional (2D) corresponding product of metamaterial, which is named metasurface, can achieve flexible manipulation of phase, amplitude, and polarization of the transmitted or reflected electromagnetic wave in the 2D plane [22,23]. Compared with a traditional photonics device or three-dimensional metamaterial, a metasurface has the characteristics of low-cost manufacturing and easy integration with other devices due to its planar structure. Therefore, the metasurface has demonstrated its advantages in applications such as anomalous refraction and reflection [24,25], spin Hall effect [26,27,28], cloaking [29], and hologram [30,31]. Currently, there are many studies on how to generate OVBs using metasurfaces [22,32,33,34], but most of them are to generate a single OVB. In order to increase the practicability and broaden the application scenarios, it is particularly critical to improve the metasurface of a single function into a multifunctional structure.

In the field of optical micro-manipulation, measuring TCs, shaping OVBs, and regulating the distribution of singularities are pivotal. Traditional methods of measuring TCs are mainly based on interference and diffraction [35]. Since constraining by their self-focusing regions, these methods greatly reduce flexibility. In addition, the manipulated particles move with the trend of energy flow, and the direction of energy flow in OVB follows the trajectory of the beam. Using this property, changing the angle between the sides of the polygon can well dominate the motion of the particles. In order to realize the control of the movement path of the particles, some progress has been made in research of using metasurfaces to accomplish OVBs shaping [36,37,38]. However, there are a few ways to achieve these two functions at the same time. Thence, in order to further increase the flexibility of OVBs, we consider adding the control of the cross-phase (CP) to the general vortex beams. CP is an extraordinary phase that can rotate both spectral density and coherence during propagation [39]. Furthermore, CP can be divided into lower-order cross-phase (LOCP) and higher-order cross-phase (HOCP) according to its order. The rational introduction of CP into the vortex beam can realize many practical functions such as determining the topological charge number and integer OVB. In 2019, Shen et al. introduced the novel phase structure LOCP into OVB for the first time [40]. They realized the measurement of the TCs of the optical vortex with torsion phase, which broke new ground in the generation and measurement of optical vortices. Then, Ren et al. added the regulation of HOCP to the perfect optical vortex (POV) and successfully reshaped the circular OVB into an arbitrary polygon vortex beam in 2020 [41]. Present studies are still realized by using large-size devices, and there has not been any research on the application of the combination of CP and OVB with a metasurface.

In this paper, we proposed and designed a novel optical vortex beam generator via cross-phase based on a metasurface utilizing numerical simulations, as shown in Figure 1. The metasurface is based on the propagation phase, using TiO_2_ square columns with high transmittance in the telecommunication waveband as unit cells, which can well implement the two functions of determining the TCs number when LOCP is incorporated and OVB shaping when HOCP is added. Moreover, this OVB generator is applicable for any number of TCs. We additionally combined the rotation component to make the generated beam rotation adjustable. The location of all functions implemented by this generator is selective. Finally, by analyzing the influence of the conversion rate *u* on the regulation of OVB by LOCP and HOCP, it is found that the metasurface can achieve singularity manipulation under the control of *u*. Numerical simulations can show that the functional diversity of traditional OVB beam can be greatly improved after adding the modulation of the CP. This is an update iteration for previous studies on general vortices [42]. Moreover, compared with the use of general optical fiber transmission, the use of the metasurface satisfies the two advantages of reducing occupied space and energy loss at the same time [43]. The designed multifunctional OVB generator has good application prospects in photon manipulation, optical communication, and vortex beam modulation due to its small size and simple structure.

## 2. Methods and Materials

Figure 2 takes the 4th-order OVB as an example to show the generation process of the phase mask composed of a focusing lens phase, vortex phase, and CP. As shown in Figure 2a, *φ*_1_(*x*, *y*) represents the focusing lens phase, whose function is to focus the vortex beam on the required focal plane along the radial direction. The spiral phase *φ*_2_(*x*, *y*) along the direction of the azimuth that produces the OAM, is obtained in Figure 2b. The phase of general focused OVB is *φ*_1_(*x*, *y*) + *φ*_2_(*x*, *y*). Then, the results of participating in LOCP on OVB are shown in Figure 2d. The result of adding HOCP regulation on OVB is shown in Figure 2f. We take 4th-order HOCP as an example to illustrate that when the HOCP modulation is added to the OVB, its phase will be modulated in a square spiral distribution, so the output polygonal OVB can be realized.

The form of the cross-phase vortex beam (CPVB) *φ*(*x*, *y*) in Cartesian coordinates (*x*, *y*) is:(1)$$\varphi (x,y)=\varphi_{1}(x,y)+\varphi_{2}(x,y)+\varphi_{3}(x,y)+n\cdot 2\pi$$
(2)$$\varphi_{1}(x,y)=-\frac{2\pi }{\lambda }(\sqrt{x^{2}+y^{2}+f^{2}}-f)$$
(3)$$\varphi_{2}(x,y)=m\cdot \arctan(\frac{y}{x})$$
(4)$$\varphi_{3}(x,y)=u\cdot (x^{p}\cdot \cos\theta -y^{q}\cdot \sin\theta )\cdot (x^{p}\cdot \sin\theta +y^{q}\cdot \cos\theta )$$
where *λ* represents the wavelength of the incident light, *f* represents the focal length of the focusing lens, *m* represents the value of TCs, the parameter *u* controls the conversion rate, the azimuth factor *θ* represents the rotation angle of the transformed beam on the Fourier plane, and *n* is any real number. The exponents *p* and *q* are any positive integers, and the sum of *p* and *q* is the order of CP. *φ*_3_(*x*, *y*) is the expression of LOCP [40] when *p* and *q* are both equal to 1 (the sum is 2), and it is the expression of HOCP [41] when the sum of *p* and *q* is greater than 2.

Based on the above theories, the schematic diagram of the OVB generator is shown in Figure 1, and the unit cell diagram is given in Figure 3. Considering the relationship between structural simplicity and efficiency, we pick the square columns arranged by the propagation phase period as the simulated structure. The unit cell is a TiO_2_ square column and the substrate is set to SiO_2_, as shown in Figure 3a. The whole metasurface includes 40 × 40 structural units in 41 μm × 41 μm area. The refractive index parameters (*n, k*) of the TiO_2_ material are taken from Ref. [44]. The real part (*n*) and imaginary part (*k*) of the refractive index are plotted in Figure 3b. In the range of visible to near-infrared light, TiO_2_ material has a relatively high refractive index. To further explore the mechanism of phase realization, we use the three-dimensional finite difference time domain (FDTD) solver to numerically simulate the structure. The size of the mesh cell we set is 0.02 µm. We choose *λ* = 1550 nm as the simulation wavelength. At this wavelength, the refractive indices of TiO_2_ and SiO_2_ are $n_{TiO_{2}}=2.05$ and $n_{SiO_{2}}=1.48$, and the imaginary part (*k*) of the refractive indexes are approximately zero. In the design, we set H = 800 nm, P = 1020 nm, and *W* changing from 100 to 1000 nm. Figure 3c shows the relationship between the effective index of a single TiO_2_ square column and its width for x-polarized incidence. The result denotes that as *W* increases, the effective refractive index gradually approaches the actual refractive index of TiO_2_, that is $n_{eff}\;\approx \;n_{TiO_{2}}$. Furthermore, it can be drawn from Figure 3d that high transmittance and phase coverage from 0 to 2π can be achieved well under this circumstance.

## 3. Results and Discussions

### 3.1. The Propagation Characteristics and the Rotation Adjustability of the LOCP

First of all, we introduce the propagation characteristics of a metasurface controlled by LOCP. We take the situation of *θ* = 0 into consideration and set *p* = *q* = 1. The CP formula can be expressed as:(5)$$\varphi_{3}(x,y)=u\cdot x\cdot y$$

We choose *f* = 40 μm as the focal length and simulated the electric field distribution from *z* = −10 μm to *z* = −100 μm under the incidence of left-handed circularly polarized (LCP) light, as shown in Figure 4. By observing the phase (see Figure 4a) and light field (see Figure 4b) distributions at different positions during the propagation process, it is found that after LOCP is introduced into OVB, the beam propagates in the form of elliptically polarized light. When it comes to the selected focal plane, a light field similar to the transverse mode can be generated to determine its TCs (see Figure 4b_3_), and finally, it restores to elliptically polarized light. At the same time, the phase will be reversed on the focal plane, namely, the −4th order reverse vortex beam is changed back to the +4th order vortex beam. This is because the CP introduces astigmatism to the OVB, which causes the mode changing of light field during the propagation [45]. It can be clearly seen from Figure 4c that this metasurface has the same self-focusing property as a traditional focused vortex beam. Compared with the traditional TCs measurement, we can realize the advantages of miniaturization and space saving by using the metasurface.

The light intensity distribution diagrams corresponding to different TCs are shown in Figure 5. The value of TC is determined by the number of bright stripes. When the TC is 0, the beam will be shaped into a large bright spot in the center, which is consistent with the result in Figure 5b. Moreover, as shown in Figure 5f, in the case where TCs are 10, the result means that the LOCP can be adopted to measure the large TCs value in OVB. LOCP can not only test the value of TCs but also determine the symbol of its TCs. By comparing Figure 5a,e, it can be found that when the TCs are negative, the direction of the light field is along the *x*-axis direction, and when the TCs are positive, it is along the *y*-axis direction. Our method can achieve in situ measurement of TCs, which is more flexible and effective than far-field measurement. It has a great effect on the development of optical control.

Next, we add the rotation component *R*(θ) to the LOCP in Equation (5):(6)$$R\left(\theta \right)=(\begin{array}{l} \cos\theta \\ \sin\theta \end{array}\begin{array}{l} -\sin\theta \\ \cos\theta \end{array}).$$

The LOCP expression with tunable rotation angle can be obtained as:(7)$$\varphi_{3}(x,y,\theta )=u\cdot (x\cdot \cos\theta -y\cdot \sin\theta )\cdot (x\sin\theta +y\cos\theta ).$$

The results of increasing the rotation angle from 0 to 0.5π are shown in Figure 6. With the increase in the rotation angle *θ*, the image formed by the propagating beam on the focal plane will rotate with the corresponding angle. Moreover, the light intensity and distribution will not change during the rotation.

### 3.2. The Propagation Characteristics of the HOCP and the Shaping Arbitrary Polygon Vortex Beams

Furthermore, we consider adding HOCP modulation to OVB, where the phase component of HOCP is given in Equation (4). Figure 7 shows the propagation characteristics of the +4th order (*m* = 4) OVB adding the HOCP with *p* = *q* = 2 (the sum is 4). By analyzing the phase (see Figure 7a) and light field (see Figure 7b) distributions in the specific XOY plane, it can be found that the propagation properties of HOCP have many similarities with LOCP; for example, both of them produce a phase flip on the focal plane. As *z* gets closer to the focus position, the singularities are more concentrated. In addition, it can be seen from Figure 7b_3_ that after the 4th-order HOCP is incorporated, the shape of the OVB is modulated into a quadrilateral in the self-focusing area. The generated beam still has good self-focusing properties. The OVB modulated by HOCP can adjust the trajectory of the controlled particles arbitrarily by controlling the energy flow distribution in the self-focusing area, which has far-reaching significance for particle manipulation.

We also analyze the influence of different values of *p* and *q* on the shaping of the OVB. Figure 8a,8b correspond to the cases where the TC is 2 and 4, respectively. By comparison, it can be seen intuitively that increasing the TCs will expand the waist radius of the polygonal beams, which is consistent with the traditional focused OVBs. Theoretically, HOCP can not only shape OVB into a quadrilateral beam but also realize any polygonal beam, as long as the value of *u* and the sum of *p* and *q* are adjusted reasonably. Through observing Figure 8, it is surprising to find that the number of any polygon beam’s sides is always equal to the sum of *p* and *q*. In other words, when the order of HOCP is 3, 4, and 5, the OVB will be shaped into triangle, quadrilateral, and pentagon, respectively. Additionally, when the values of *q* and *p* are inconsistent, the symmetry of the generated polygonal beam will also change, which can be found by comparing Figure 8b_4_ with Figure 8b_3_. The greater the difference between the *p* and *q* values, the more severe the distortion of the generated polygonal beam. Thus, the trajectory of the manipulated particles can be accurately determined, provided that the values of *p* and *q* can be changed according to the required results.

### 3.3. The Influence of Self-Selected Area and Conversion Rate on OVB Generator

We illustrate the influence of the self-selected area on the realization of the primitive functions. Figure 9a,9b represent the LOCP; Figure 9c,9d represent the HOCP. We set the focal wavelengths as *f* = 20 μm, *f* = 30 μm, *f* = 40 μm, *f* = 50 μm, and *f* = 60 μm, respectively. It can be found that corresponding to LOCP and HOCP, the metasurface can perform their original functions well on different focal planes. As the focusing wavelength shifts to the right, its ring radius and the realization range will increase. Therefore, it confirms that the metasurface can well realize the functions of determining TCs and shaping OVB in the self-selected area, which can improve the flexibility of optical particle manipulation.

Finally, we specifically show the impact of the conversion rate *u* on LOCP and HOCP. The images of light field distribution correspond to Figure 10a,10c, and the images of phase distribution correspond to Figure 10b,10d, respectively. Using Equation (4) to calculate, the value range of *u* should be between 10^10^ and 10^11^. Through numerical simulation, when the *u* value is relatively smaller, the reduction in conversion rate makes the OVB inseparable from the original circular motion trajectory, which in turn causes the adhesion phenomenon between each bright spots, and the value of the TCs cannot be determined. However, if the conversion rate *u* is too high, the OVB will be excessively separated. The surrounding secondary light sources will interfere with the main working beam, thereby reducing the conversion efficiency. In this case, the OVB incorporated with LOCP control is not stable enough. Then, similar to the way of analyzing LOCP, we get the magnitude of *u*, which should be 10^20^. When the *u* value is less, a polygonal beam with a complete closed loop can still be realized on the focal plane. However, as the value of *u* gradually increases until it exceeds the cut-off value, the beam generated at the focal plane will split due to the excessively high conversion rate. Moreover, by observing Figure 10d, it can be further found that as the *u* value increases, the distance between each singularity of OVB will become farther. Therefore, the singularities of OVB can be well regulated by adjusting the value of *u*, which has great value in the field of multi-particle manipulation.

## 4. Conclusions

To sum up, we have proved the different functions of the multifunctional OVB generator regulated by CP through simulation. When LOCP is added into OVB, TCs can be determined by observing the number of bright and dark stripes. The dispersion effect, rotation angle, and TCs quantity of OVB can be adjusted by changing the parameters of *u*, *θ*, and *m*, respectively. When OVB is adjusted by HOCP, it can realize shaping the ordinary OVB into a special polygonal beam. The values of *p* and *q* can be changed according to the desired result to complete the generation of different polygon beams, so as to achieve the goal of arbitrary photon beam rectification. The results produced by the generator can be adjusted in the self-selected area through adjusting *f*. Finally, it is found that as the value of *u* increases, the singularity positions of OVB are more dispersed. This method does not require complex interferometer settings or calculation processes. Consequently, this OVB generator may have good application prospects in the optical field in terms of photon manipulation, quantum information processing, and vortex beam modulation in the future.

## Figures and Tables

**Figure 1 nanomaterials-12-00653-f001:**
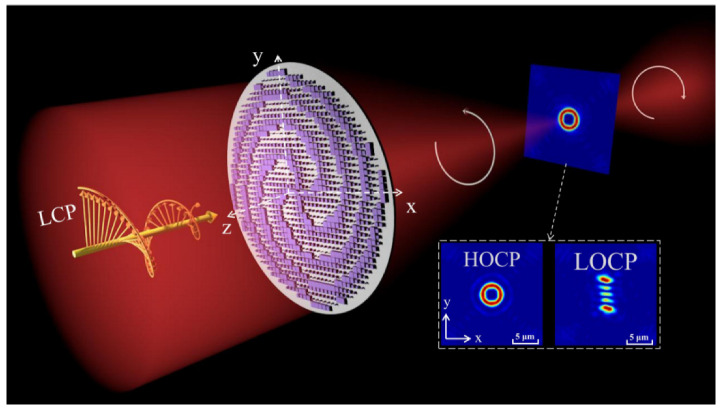
Schematic diagram of the OVB generator combined with cross-phase based on metasurface.

**Figure 2 nanomaterials-12-00653-f002:**
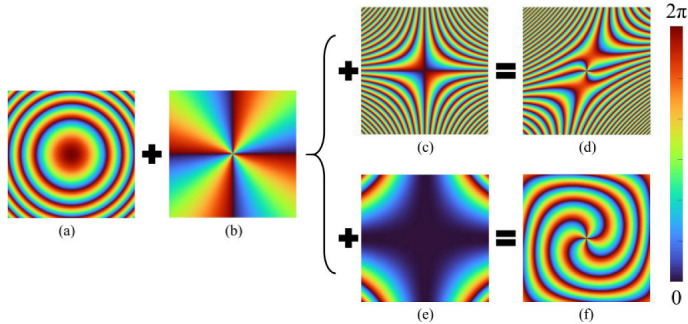
Phase mask generation process. (**a**) Focused lens phase. (**b**) Optical vortex phase. (**c**) LOCP phase. (**d**) The generated LOCP phase mask. (**e**) HOCP phase. (**f**) The generated HOCP phase mask.

**Figure 3 nanomaterials-12-00653-f003:**
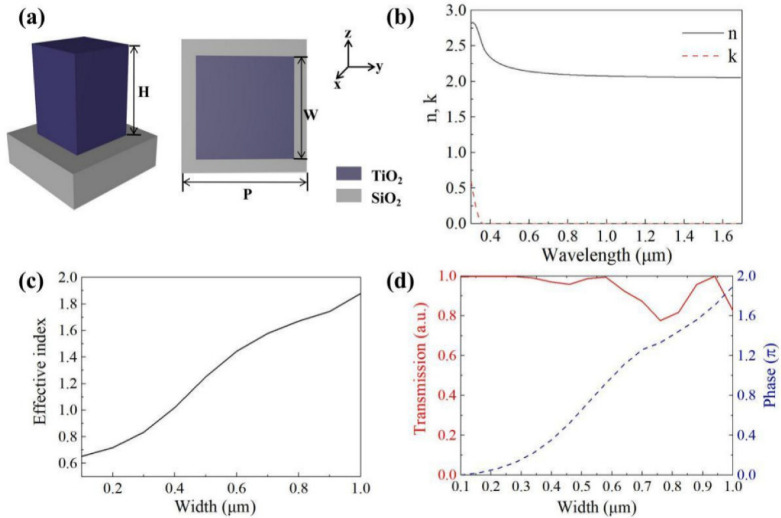
(**a**) Unit cell schematic diagram. (**b**) TiO_2_ refractive index parameters (*n*, *k*). (**c**) The effective refractive index of a square TiO_2_ column under 1550 nm incident light. (**d**) The transmission efficiency and phase change when *W* is increased from 100 to 1000 nm.

**Figure 4 nanomaterials-12-00653-f004:**
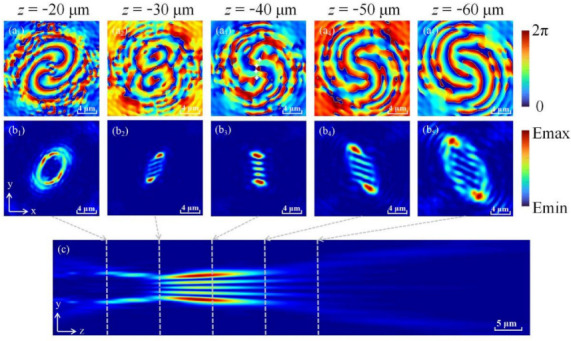
Schematic diagram of beam propagation with LOCP. The phase distribution (**a**) and the light field (**b**) diagrams of the XOY plane from *z* = −20 μm to *z* = −60 μm. (**c**) The optical field distribution diagram of the LOCP beam in the YOZ plane.

**Figure 5 nanomaterials-12-00653-f005:**
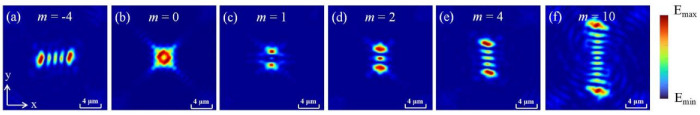
Optical field distribution of focused vortex beam based on LOCP under different TCs, where (**a**) corresponds to *m* = −4, (**b**) corresponds to *m* = 0, (**c**) corresponds to *m* = 1, (**d**) corresponds to *m* = 2, (**e**) corresponds to *m* = 4, and (**f**) corresponds to *m* = 10.

**Figure 6 nanomaterials-12-00653-f006:**
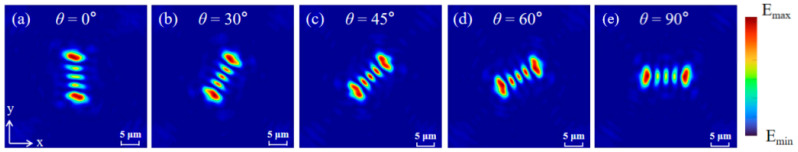
The optical field distribution of the focused vortex beam under the rotation angle *θ* based on LOCP, where (**a**) corresponds to *θ* = 0°, (**b**) corresponds to *θ* = 30°, (**c**) corresponds to *θ* = 45°, (**d**) corresponds to *θ* = 60°, and (**e**) corresponds to *θ* = 90°.

**Figure 7 nanomaterials-12-00653-f007:**
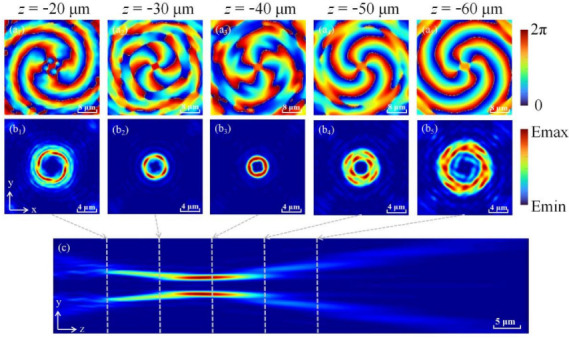
Schematic diagram of beam propagation with HOCP. The phase distribution (**a**) and the light field (**b**) diagrams of the XOY plane from *z* = −20 μm to *z* = −60 μm. (**c**) The optical field distribution diagram of the HOCP beam in the YOZ plane.

**Figure 8 nanomaterials-12-00653-f008:**
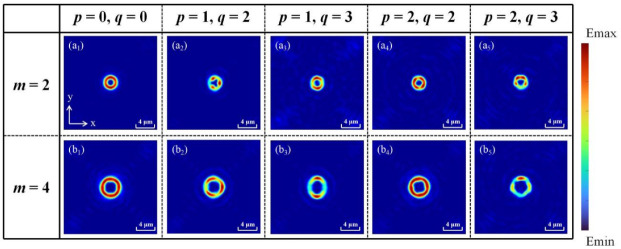
Polygonal beams generated by HOCP with different values of *p* and *q* when TCs are (**a**) *m* = 2 and (**b**) *m* = 4. (**a_1_**,**b_1_**) correspond to ordinary vortex beams. (**a_2_**,**b_2_**) correspond to the 3rd HOCP. (**a_3_**,**b_3_**) correspond to the special 4th HOCP. (**a_4_**,**b_4_**) correspond to the 4th HOCP. (**a_5_**,**b_5_**) correspond to the 5th HOCP.

**Figure 9 nanomaterials-12-00653-f009:**
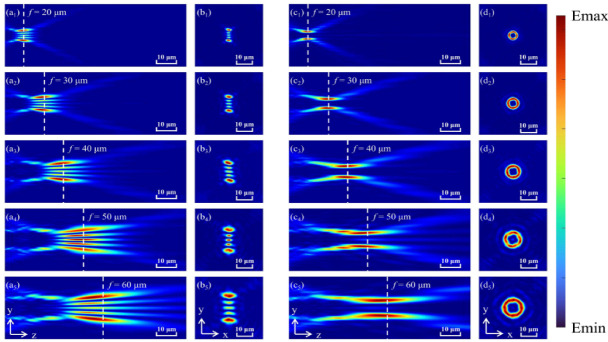
The influence of different focal wavelengths on the OVB generators. (**a**) corresponds to the LOCP of the YOZ plane. (**b**) corresponds to the LOCP of the YOX plane. (**c**) corresponds to the HOCP of the YOZ plane. (**d**) corresponds to the HOCP of the YOX plane.

**Figure 10 nanomaterials-12-00653-f010:**
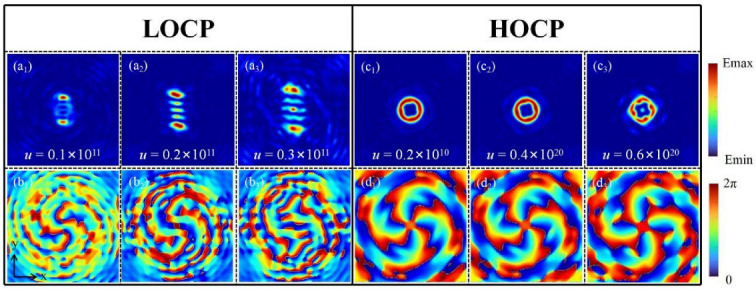
The influence of different *u* values on LOCP and HOCP. (**a**) LOCP light field distribution. (**b**) LOCP phase distribution. (**c**) HOCP light field distribution. (**d**) HOCP phase distribution.

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
