# Peer review of "Multifunctional Optical Vortex Beam Generator via Cross-Phase Based on Metasurface"

_nanomaterials, 2022, doi:10.3390/nano12040653_

Round 1
Reviewer 1 Report
In the article entitled “Multifunctional optical vortex beam generator via cross-phase based on metasurface”, the authors describe a multifunctional optical vortex beam (OVB) generator via cross-10 phase based on metasurface.
The focus is on the results of simulation (or experiment?) and a brief description of the underlying physical mechanisms. The manuscript is rather well organized, the results are mostly well described.
The conclusions are supported by evidence, however, it is not clear if the evidence is experimental or purely numerical.
The innovative aspect of the paper is relatively limited. In general, some aspects should also be discussed in more detail to improve clarity. More detailed remarks are given below. After making satisfactory revisions I would recommend the manuscript for publication in Crystals.
- The introduction is written in a very chaotic way, especially the beginning - jumping rapidly from topic to topic without any substantive connections.
- The introduction lacks information about LOCP and HOCP, what exactly they are and why they were introduced. Only in the next part, it is mentioned why they are introduced, but this explanation was missing earlier. There are of course literature references, but a few more sentences of description would improve the reading comfort and understanding of the issue.
- Is the only element of novelty the addition of an element that modifies the vortex phase? This is not clearly emphasized.
- The paragraph in lines 207-222 - in my opinion, is not convincing enough. While most of the results from Fig. 8 actually confirm the formation of p+q polygon-shaped beams, Fig. a3,b3 completely shows this, even giving this correction that for p!=q the vortex will be disturbed. The last sentence of the paragraph does not connect to the previous sentences - suddenly there is talk of manipulated particles, which only appear for the first time in the text without even a sentence of the introduction.
- I feel that there is no separation of discontinuities in the HOCP beams in Fig. 10d described in 258-261
- There was no clear information about whether at least one such structure was prepared or only simulated.
- Generally from the presented results, one can conclude that the proposed structure works, however some of the results and especially their presentation need to be tightened and clarified.
Reviewer 2 Report
The Authors discuss a system in which optical vortices could be generated. In particular, they propose a multifunctional optical vortex beam (OVB) generator system with a cross-phase. The device's design is based on a metasurface built in the form of TiO2 square columns characterized by a high transmittance in the telecommunication waveband. Additionally, the device contains the rotation component, making the generated beam rotation adjustable. The Authors have determined the propagation characteristics of the device. Discussing the low- and high-order cross-phase cases, they have shown that singularity manipulation can be achieved under the control of the conversion rate. They indicate that the proposed OVB design is promising in the context of its application in optical communication, vortex beam modulation, and finally, in photon manipulation.
The presented design and the discussion of the results seem to be interesting enough to deserve publication, and they are especially valid from the practical point of view. Presented calculations seem to be correct, and the results are shown and discussed in an understandable way, making them easily accessible to the Readers. Thus, I recommend the manuscript for publication in its present form.
Reviewer 3 Report
In this paper an optical vortex beam generator is presented with metasurface structure. The concept appears interesting for different frequency band (microwave, millimeter-wave and above). The author should clearly state what is the advantage of their design in comparison to other known in literature. The expression of the paper is generally appropriate, but needs some minor modifications. The comments are listed as follows:
Comments for Authors:
- The abstract should be updated, include the quantitative result of the work.
- Lack of comparison of obtained results with similar research. It is recommended, authors do a more comparative study between proposed OAM generator and other works in a comparison table (clear the work benefits).
- Refer to similar vortex beam generator, in introduction, with different technique, such as:
[1] Mutual coupling reduction using plane spiral orbital angular momentum electromagnetic wave, Journal of Electromagnetic Waves and Applications, 2022.
[2] Optical-controlled fast switching of radio frequency orbital angular momentum beams with different mode and radiation direction, Journal of Lightwave Technology, 2022.
[3] High-directivity orbital angular momentum antenna for millimeter-wave wireless communications, IEEE Transactions on Antennas and Propagation, 2021.
